# Anion Exchange Membrane Based on Sulfonated Poly (Styrene-Ethylene-Butylene-Styrene) Copolymers

**DOI:** 10.3390/polym13101669

**Published:** 2021-05-20

**Authors:** Hye-Seon Park, Chang-Kook Hong

**Affiliations:** Polymer Energy Materials Laboratory, School of Chemical Engineering, Chonnam National University, Gwangju 61186, Korea; parkhseon17@gmail.com

**Keywords:** SEBS, membrane, maleic anhydride, water uptake, impedance spectroscopy, ionic conductivity

## Abstract

Sulfonated poly(styrene-ethylene-butylene-styrene) copolymer (S-SEBS) was prepared as an anion exchange membrane using the casting method. The prepared S-SEBS was further modified with sulfonic acid groups and grafted with maleic anhydride (MA) to improve the ionic conducting properties. The prepared MA-grafted S-SEBS (S-SEBS-g-MA) membranes were characterized by Fourier transform infrared red (FT-IR) spectroscopy and dynamic modulus analysis (DMA). The morphology of the S-SEBS and S-SEBS-g-MA was investigated using atomic force microscopy (AFM) analysis. The modified membranes formed ionic channels by means of association with the sulfonate group and carboxyl group in the SEBS. The electrochemical properties of the modified SEBS membranes, such as water uptake capability, impedance spectroscopy, ionic conductivity, and ionic exchange capacity (IEC), were also measured. The electrochemical analysis revealed that the S-SEBS-g-MA anion exchange membrane showed ionic conductivity of 0.25 S/cm at 100% relative humidity, with 72.5% water uptake capacity. Interestingly, we did not observe any changes in their mechanical and chemical properties, which revealed the robustness of the modified SEBS membrane.

## 1. Introduction

The use of renewable energy sources, such as proton exchange membrane fuel cells (PEMFC), is currently growing significantly owing to their environmental, social and economic benefits [1,2]. The conventional renewable energy sources, such as solar and wind, are intermittent and often unpredictable due to their dependency on the weather conditions. These characteristics limit the degree to which utilities can rely upon them, and currently such renewable energy alternatives comprise a small percentage of the primary power sources on the electrical grid [3]. The need to satisfy the energy demand during periods of low energy production has inspired the development of efficient energy storage systems. Recently, the redox flow battery (RFB) has attracted an enormous amount of attention as a promising energy storage system for solar power, nuclear power, emergency uninterruptible power supply (UPS), and for batteries in electric vehicles. In particular, large-scale storage systems are more beneficial as they offer a guaranteed energy supply when using renewable energy sources over a long period. RFBs present several advantages that makes them promising candidates for large-scale energy storage systems; they have energy and power density, capacity which can be designed independently and easily modified even after installation, a moderate operational temperature and a long-life, which makes them highly reliable [3,4].

An RFB consists of electrolyte tanks from which the oxidant and reductant electrolytes are circulated by pumps, through a cell stack comprising a number of connected cells. Each cell comprises an anode and a cathode, which are separated by an ion exchange membrane (IEM). The exchange membrane has been widely used in many fields of life. For example, PEMFC, which includes a membrane, and two electrodes, has grown up with huge attraction because of its simple operation and fuel availability. The PEMFCs have attracted attention from energy devices such as portable, mobile and stationary devices, since it helps in the effective reduction in energy shortage and environmental pollution. The IEM will play an important role in the future of electrical energy generation, which is considered as renewable and clean energy. In addition, the IEM facilitates an improved conductive path between the electrolytes. Therefore, it acts as a membrane separator by preventing cross-mixing and direct chemical reaction of the oxidant and reductant electrolytes from the two reservoir tanks [5,6]. Notably, the ion exchange membrane is a major determinant of redox flow batteries’ performance, highlighting the importance of an ideal membrane [7].

Perhaps the most commercially advanced RFB industry that uses a proton-exchange membrane is the vanadium redox-flow battery (VRFB) system [8]. An ion exchange film is an ionic membrane that can selectively separate cations and anions and is an optional transmission film with a functional group that can attract or repel the ions. The IEM has selective permeability to counter-ions due to fixed ions in the actuator. This means that the charged agonist is fixed to the membrane and selectively transmits only the counter-ion with different charges than the agonist, and not the co-ion with the same charge as the agonist. This is called the Donnan exclusion effect—that is, the cation exchange film selectively permeates the cation, and the anion exchange film selectively permeates the anion. The Donnan exclusion effect also causes an unbalanced electrochemical potential difference between the electrolyte and ion exchange membranes, resulting in potential differences in the boundary of the IEMs. This potential difference causes the ions to move until both electrochemical potentials reach equilibrium, which is called the Donnan equilibrium [9]. The most widely used polymer electrolyte membrane is the Nafion 117, a perfluorinated cation exchange film developed by Dupont in 1968. Perfluorometer ion exchange membranes have excellent ion conductivity, chemical stability, and dimensional stability, but crossover occurs due to low ion selection, and above all, they are expensive. To overcome these shortcomings, many studies are being conducted on the development of low-cost hydrocarbon ion exchange membranes [10]. Another important reason for the IEM study is that ion exchange capacity (IEC) will play an important role in the future of electrical energy generation, which is considered as renewable and clean energy [8].

Earlier studies clearly demonstrated that the block copolymers provide excellent separation properties [11,12,13,14,15]. The poly(styrene-ethylene-butylene-styrene) copolymer (SEBS) is one of the promising materials for membrane separators due to its high thermal, chemical, and tunable mechanical properties, and cost effectiveness. The SEBS tri-block polymer has further attracted considerable interest because of its promising proton conducting properties [16,17]. Moreover, due to its simple structure over well-known Nafion 117, the SEBS is considered as a promising anion exchange membrane. It is well known that the key mechanical and electrochemical properties can be tuned via composition of the backbone, hydrocarbon versus fluorocarbons [17,18]. Generally, sulfonated copolymers are synthesized either by a direct copolymerization method or post sulfonated technique [19]. Recently, Mohanty et al. functionalized SEBS membranes through borylation using Suzuki coupling reactions and demonstrated IECs of ca. 2.2 mmol/g [20]. However, the maleic anhydride (MA)-grafted SEBS membrane not yet studied. The MA has ability to enhance the electrochemical properties of SEBS membrane due to its promising solubility properties.

Moreover, while many studies have been conducted on IEM with sulfonation of SESB, the ionic conductivity behavior between sulfonyl and carboxyl functional group in the ionic exchange membrane has not been carried out in previous studies. In this study, we investigated the ionic conductivity behavior of the SEBS membrane with the help of different sulfonyl and carboxylic groups. We synthesized a high-quality sulfonated SEBS-grafted-MA membrane (S-SEBS-g-MA), which permits higher conductivity than conventional Nafion 117, and exhibits a good mechanochemical property with the help of sulfosuccinic acid by cross-linking mechanism. Our electrochemical analysis revealed that the modified membrane shows improved proton conductivity, IEC and water uptake properties. In addition, the dynamic modulus analysis (DMA) result confirms the improved modulus properties.

## 2. Materials and Methods

### 2.1. Materials

The polymeric material used in this experiment was poly(styrene-ethylene-butylene-styrene) copolymer (SEBS, ~118,000 g/mol, 28 wt% polystyrene, Sigma Aldrich, Seoul, Korea). Maleic anhydride (MA, 98.96 g/mol, Daejung, Siheung, Korea) was used as the grafting agent. Chloroform (99.8%, Aldrich, Seoul, Korea) and ρ-xylene (99%, Sigma Aldrich, Seoul, Korea) were used as a solvent, while dicumyl peroxide (DCP, 98%, Sigma Aldrich, Seoul, Korea), sulfosuccinic acid (SSA, 70 wt%, Sigma Aldrich, Seoul, Korea) and chlorosulfonic acid (99%, Sigma Aldrich, Seoul, Korea) were used as an initiator, the crosslinking agent and sulfonation inhibitor, respectively.

### 2.2. Synthesis of S-SEBS and S-SEBS-g-MA Membrane

#### 2.2.1. Sulfonated SEBS (S-SEBS)

The S-SEBS were prepared by following procedures reported in the previous literature [21,22]. In this process, 10 wt% SEBS was dissolved completely in chloroform by being vigorous stirred for 3 h. The 30 mL of above solution was casted in a circular glass Petri-dish which results in ~150 μm thickness after drying at room temperature for 12 h. The dried film was removed from the glass substrate and cut into a 1 cm × 1 cm square shape. The sulfonation agent was prepared by diluting chlorosulfonic acid in 1,2-dichloroethane and the prepared SEBS membrane was soaking into sulfonation agent for 5 min. Then, the modified membrane sample washed several times with deionized water. The membrane was immersed in deionized water over 24 h, before the tests were carried out.

#### 2.2.2. Synthesis MA Grafted S-SEBS (S-SEBS-g-MA)

We used the backbone-functionalization method for the preparation of S-SEBS-g-MA membrane. The typical scheme used for the preparation is shown in Figure 1. In typical synthesis, 10 wt% SEBS was dissolved in ρ-xylene for 3 h with continuous stirring. Then, maleic anhydride (MA) of 10 wt% was added into the SEBS solution under nitrogen gas at 135 °C and stirred for 1 h. Next, dicumyl peroxide (DCP) as an initiator in ρ-xylene was added and then stirred for another 1 h. After cooling at room temperature, 0.5 g sulfosuccinic acid (SSA) is then added in the above solution. The prepared SEBS-g-MA solution was casted on a glass substrate with a thickness of approximately 150 μm and then dried for 12 h at room temperature. The completely dried film was carefully removed from the glass substrate and cut into a square shape (size 1 cm × 1 cm). The SEBS-g-MA membrane was soaked in sulfonation agent for 5 min, and then washing with deionized water several times. The membrane was immersed in deionized water over 24 h before the test was carried out.

### 2.3. Characterizations

#### 2.3.1. Fourier Transform Infrared (FT-IR) Spectroscopy

The FT-IR spectrometer (Spectrum 400, Perkin Elmer, Gwangju, Korea) was used to investigate the grafting and the functional groups in the synthesized membrane qualitatively.

#### 2.3.2. Gel Permeation Chromatography (GPC)

The measurements were conducted using Shodex KF-804, Shodex KF-802, and Shodex KF-801 column (HLC-8320 GPC, Tosoh, Gwangju, Korea) at 40 °C with the eluent flow rate of 1 mL/min and injection of 100 μL. All samples were dissolved in tetrahydrofuran (THF) before measurements. Polystyrene standard was used as a reference.

#### 2.3.3. Topographical Analysis

The surface topography of the membranes was investigated using atomic force microscopy (AFM, XE-100, Park system, Gwangju, Korea) operated in tapping mode. For AFM images, the SEBS-based anion exchange membrane films were prepared on the Si wafer at 2000 rpm for 30 s and dried overnight in ambient conditions.

#### 2.3.4. Ionic Conductivity

The ionic conductivity of the membrane was obtained by impedance spectroscopy measurement using a conductivity analyzer (Ivuimstat, HS Technologies, Gwangju, Korea) with the frequency range of 1 Hz to 1 MHz at room temperature. Before measurements, the prepared membrane was immersed in deionized water for over 24 h. During measurements, the contact area of the membrane between electrodes was maximized to reduce errors in ionic conductivity measurements. The experimental arrangement for ionic conductivity is shown in Figure 2. For impedance reproducibility and cross-check verification, we successively repeated samples from each set at least five times. The ionic conductivity of the membrane was calculated by using the following equation.
(1)$$\sigma \left(S/\mathrm{cm}\right)=\frac{L}{\mathrm{RA}}$$
where R is the real impedance taken at zero imaginary impedance in the impedance spectroscopy, and L and A are the thickness and area of the membrane, respectively.

#### 2.3.5. Ion Exchange Capacity (IEC)

For the ion exchange capacity (IEC) measurement, the dried sample with a certain weight was immersed into 1 M HCl solution overnight and stirred. The ion exchange capacity was determined by titrating the solution with 1 M NaOH solution. The IEC of each sample was measured at least 5 times. The IEC of the membranes was calculated by the following equation.
(2)$$\mathrm{IEC}(\mathrm{meq}/g)=\frac{M_{O\left(\mathrm{HCl}\right)}-M_{e\;\left(\mathrm{HCl}\right)}}{W_{\mathrm{dry}}}$$
where M_o(HCl)_ and M_e(HCl)_ are the milliequivalent (meq) of HCl acquired before and after the equilibrium, respectively, and W_dry_ is the mass (g) of the dried membrane.

#### 2.3.6. Sulfonation Degree (SD)

The sulfonation degree (SD) is related to the actual content of sulfonated poly-styrene group of S-SEBS. The SD was calculated using the following equation.
(3)$$\;\mathrm{SD}=\frac{M_{P}\times \mathrm{IEC}}{\left\{1000-\left(M_{f}\times \mathrm{IEC}\right)\right\}}$$
where M_p_ is the molecular weight of the non-functional polymer repeat unit (SEBS) and M_f_ is the molecular weight of the functional group (SO_3_H). The values for M_p_ and M_f_ are 18,000 and 81, respectively.

#### 2.3.7. Water Uptake

To evaluate the water uptake of the membrane, the sample with a certain weight was immersed in deionized water for 24 h at room temperature. After 24 h, the membrane was taken out and the water on the surface was quickly wiped using filter paper, and then the membrane was weighed again. The water uptake was calculated according to the following equation. The water uptake of each sample was measured at least 5 times and we used the average value.
(4)$$\mathrm{Water}\;\mathrm{uptake}\left(\%\right)=\frac{W_{\mathrm{wet}}-W_{\mathrm{dry}}}{W_{\mathrm{dry}}}\times 100$$
where W_wet_ and W_dry_ are the weight of the membrane in the wet and dry state, respectively.

#### 2.3.8. Mechanical Properties

Dynamic mechanical properties of the membranes were measured using dynamic mechanical analysis (DMA, universal V3.5B, TA instruments, Seoul, Korea) in tension mode. Samples were heated to 200 °C at frequency of 1 Hz, with a programmed heating rate of 5 °C/min.

## 3. Results and Discussion

### 3.1. Vibrational and GPC Analysis

Figure 3, containing the FT-IR spectra, shows SEBS peaks at 2916 and 2850 cm^−1^, which represents the CH_3_ and CH_2_ stretching. There is another peak at 1454 cm^−1^ that correspond to –CH_2_, and C=C. S-SEBS has two characteristic peaks at 1226 and 1041 cm^−1^ because of the S=O symmetric stretching vibration and the S=O asymmetric stretching vibration of the SO_3_H groups, respectively.

However, after grafting by MA, S-SEBS-g-MA has a new peak a at 3306 cm^−^^1^ because of the hydroxyl group together with some unassociated hydroxyl species in the region 3500–3200 cm^−^^1^. There are also several other new peaks at 1942 and 1751 cm^−^^1^ (asymmetric and symmetric carbonyl vibration of MA), 1600 cm^−^^1^ (C=O stretching of carboxyl group from anhydride), 1379 cm^−^^1^ (aldehyde), 1028 cm^−^^1^ (the deformation vibration of the C-H bond) [23,24]. The FT-IR results suggest that the functional groups are responsible for ionic transfer channel of hydroxide ions are successfully grafted on SEBS.

Changes average molecular weight (M_w_) distribution for SEBS, S-SEBS, SEBS-g-MA, S-SEBS-g-MA, and cross-linked S-SEBS-g-MA (CS-SEBS-g-MA) are presented in Table 1. The M_w_ of SEBS-g-MA, where MA is grafted onto SEBS, is slightly increased compared to SEBS. However, after sulfonization (S-SEBS, S-SEBS-g-MA), M_w_ decreased despite the addition of sulfonization compared to the pre-sulfonizaiton samples (SEBS, SEBS-g-MA). From this result, decomposition occurred in the polymer chain during the sulfonization. CS-SEBS-g-MA has the highest molecular weight because the higher molecular weight content of the test specimen increases, which is attributable to the crosslinking of the polymer chain.

### 3.2. Topographical Analysis

The triblock copolymer SEBS, consisting of hard and soft blocks that usually exhibit a phase-separated morphology, which has been widely studied by AFM techniques [25,26,27,28,29]. It is well known that, for high ionic conductivity of the membrane, a continuous network of a proton conducting phase within the material is essential. To check the surface morphology of membranes, the AFM images of the bare SEBS and modified SEBS membranes were recorded (Figure 4). In Figure 4a, SEBS membrane has a well-defined micro-phase separated morphology, where the dark regions represent the soft polyethylene (PE) block phase. On the other hand, the bright regions represent the stiff polystyrene (PS) phase. Ion exchange membranes of block copolymers consisting of hydrophilic and hydrophobic blocks associate uniformly on microscopic scales and their morphologies are aspherical, cylindrical, or lamellar in shape depending on the relative volume fractions of the constituent components [27,29]. Partially sulfonated SEBS block copolymers are known to demonstrate self-assembling phenomena, which lead to the separation of microphase domains [17,29].

Such microphase separation can result in the formation of continuous ionic channels that enable proton ions transportation through ionic channels. Flexible side chains were also considered to form microphase separation morphology [26]. As a result, we observed well-ordered and more continuously connected nano-channels. Proton transports through these nano-channels with hydrophilic domains lead to good water uptake and ion conductivity. The formation of continuous ionic channels led to proton ion transports through ionic channels. Formation of the continuous ionic channel is dependent on the hydrophilic functional groups such as SO_3_H and –COOH. By comparing the SEBS-g-MA membrane with –COOH in Figure 4b and S-SEBS membrane with SO_3_H in Figure 4c, different morphologies are observed in the AFM images. These results show that different surface chemical composition characteristics occur depending on the functional groups. Due to this different morphological behavior, ion clusters have different ion transportation speed. As a result, the ionic conductivity of membranes with a –COOH functional group is shown to be different from membranes with a SO_3_H functional group. A membrane with two kinds of hydrophobic functional group (SO_3_H, –COOH) has well-ordered and more continuously connected nano-channels than S-SEBS and SEBS-g-MA, which have single hydrophobic functional groups of SO_3_H or COOH, respectively (Figure 4d). These results show that S-SEBS-g-MA is superior to S-SEBS and SEBS-g-MA as an ionic exchange membrane.

### 3.3. Ionic Conductivity and Ion Exchange Capacity (IEC)

The ion exchange membranes contain a high concentration of the fixed ionic groups. On the other hand, the backbone of the membrane is extremely hydrophobic, whereas the charged acid groups are strongly hydrophilic and polar. The hydrophilic domains absorb water and form small clusters distributed throughout the backbone. Proton ions easily permeate the cationic membranes containing fixed negative groups [11]. The ion cluster forms with negative functional groups and the membrane with well-formed ion clusters have high ionic conductivity. Ideal membranes for an RFB should possess good chemical stability and high ionic conductivity. From the impedance results, we observed more hydrophilic functional groups increased, which generates continuous ionic channels responsible for increased ionic conductivity, and decreased resistance, Figure 5.

The resulting values of ionic conductivity and impedance are given in Table 2. As expected, the ionic conductivity is enhanced by increasing the concentration of hydrophilic functional groups. The S-SEBS and S-SEBS-g-MA exhibit improvement in ionic conductivity. The ionic conductivity of the S-SEBS-g-MA increases with sulfonated time and reaches a maximum value 0.18 S/cm at room temperature. As sulfonated time increases, the ionic channels also increase, resulting in an increase in the ionic conductivity. Compared with the ionic conductivity of the S-SEBS membrane (0.1 S/cm), S-SEBS-g-MA shows much higher ionic conductivity.

This is because S-SEBS has only SO_3_H, but S-SEBS-g-MA has both SO_3_H and COOH_,_ so S-SEBS-g-MA has many more hydrophilic functional groups than S-SEBS. We also used a crosslinking agent containing hydrophilic group to improve both mechanical properties and ion conductivity. As a result, the CS-SEBS-g-MA, which contains more hydrophilic ionic channels than the S-SEBS-g-MA, has an ionic conductivity of 0.25 S/cm. This also indicates that the hydrophilic groups increased the ionic conductivity. Similar to ionic conductivity IEC, the membranes are strongly dependent on the amount of the hydrophilic functional groups. The presence of water plays a vital role in ionic conductivity characteristics; its presence inside the ion exchange membranes offers transport channels for ions. Hence, higher water content will shorten the ion movement pathway and result in higher ionic conductivity. In Table 2 and Figure 6, the IEC value of the S-SEBS membrane is 2.1 meq/g which is much higher than the commercial Nafion 117 membrane, whose IEC is only 0.9 meq/g (Table 2). This means that the IEC also increases as the hydrophilic functional groups increases.

### 3.4. Water Uptake

Next, we investigated the water uptake behavior of all of the above membranes. Figure 7 shows the water uptake of the Nafion 117, S-SEBS, SEBS-g-MA, S-SEBS-g-MA and CS-SEBS-g-MA membrane. Water uptake by the membrane depicts the hydrophobicity of the membrane. The COOH activity in the membrane increases affinity for water more than SO_3_H, which affected the SEBS-g-MA membrane, causing it to swell more than S-SEBS membrane. The water uptake value for the S-SEBS membrane containing SO_3_H functional group is 33% and of the SEBS-g-MA membrane containing COOH functional group is 42% (Figure 7). Thus, S-SEBS exhibits higher ionic conductivity than the SEBS-g-MA membrane, because of ionic conductivity depending on the hydrophilic functional group and the main structure material. Although membranes have identical functional groups, they have differential water uptake values. This is because the water uptake of the membrane is related to the number of hydrophilic groups in the membrane. In other words, the water uptake increases with the increase in the hydrophilic portion of the functional groups in the membrane.

In comparison with SEBS-g-MA and S-SEBS-g-MA, the S-SEBS-g-MA shows a much higher water uptake than the SEBS-g-MA. The SEBS-g-MA contains only a hydrophilic COOH functional group, while S-SEBS-g-MA comprises two hydrophilic functional groups of SO_3_H and COOH. Thus, the higher water uptake of the S-SEBS-g-MA is due to it having more hydrophilic groups. To increase the ionic conductivity of membrane, we used the SSA as a crosslinking agent. SSA has sulfonic acid and carboxylic groups, so SSA play dual roles in the membranes; ion clustering of the hydrophilic group and crosslinking agent. By using the SSA, we increased the level of hydrophilic functionality and the water uptake of CS-SEBS-g-MA to higher than that of S-SEBS-g-MA. Following the results between ionic conductivity and water uptake, we can expect that investigating the functional groups such as sulfonyl and carboxylic acid of the membrane leads to increased ion conductivity. The modified membrane CS-SEBS-g-MA with sulfonyl and carboxylic acid functional groups leads to improved charge density of the modified membrane. From conductivity analysis, we observed significant improvement in the ionic conductivity from 0.1 to 0.18 S/cm, respectively, for the bare S-SEBS to S-SEBS-g-MA membrane with increasing functional groups. Interestingly, this conductivity reached up to 0.25 S/cm for CS-SEBS-g-MA membrane. This clearly indicates that water uptake and ionic conductivity are completely dependent upon the functional groups and cross-linking. This gives rise to the question of how ion conductivity affects the water uptake behavior of the membrane. Increasing the number of functional groups is directly proportional to increasing the ability of H^+^ transport in the membrane, which leads to improved ionic conductivity and water uptake.

### 3.5. Dynamic Mechanical Analysis (DMA)

A general problem of homogeneous sulfonated styrene main-chain polymers is that these ionomers begin to swell to strong and thus lose their mechanical stability when reaching a certain sulfonated degree. Therefore, reducing the swelling degree of the membranes without lowering their proton conductivity too much is required. These requirements are achieved by cross-linking of the ionomer membrane [30]. Flexible ionomer networks can be built up via ionic crosslinking which contain ionic crosslinks formed by proton transfer. By adding elements with relatively strong electrical voice such as SOH_3_ and -COOH, it is possible to synthesize colorless and transparent polymer materials by forming an undetermined curved chain structure.

The hydroperoxide sites at the macromolecules can cause radical chain scission [31]. C–H bond of polystyrene is easily attacked by oxygen, forming hydroperoxide radicals [30]. The hydroperoxide sites at the macromolecules can cause radical chain scission. Ionomer crosslinked membranes show reduced brittleness when dried out, compared to uncross-linked or covalently cross-linked ionomer membranes, which is possibly caused by the flexibleness of ionic cross-links [31]. We used the SSA which contains two types of hydrophilic groups, i.e., SO_3_H and COOH as the crosslink agent. The use of the SSA results in a higher ionic conductivity and improved physical properties of the ion exchange membrane. We compared the storage modulus before and after crosslinking of the ion exchange membrane. We observed that the storage modulus of the S-SEBS and S-SEBS-g-MA showed similar behavior, but the storage modulus of the CS-SEBS-g-MA increased by crosslinking between the molecular chains of S-SEBS-g-MA and SSA (Figure 8). The CS-SEBS-g-MA membrane exhibited 50% higher storage modulus than the SEBS-g-MA membrane. These results show that SSA is a good crosslinking agent for the SEBS-g-MA to enhance the mechanical properties and ion conductivity.

## 4. Conclusions

In conclusion, we successfully prepared the ionic exchange membrane from SEBS via sulfonation, grafting MA and crosslinking. This study showed that the ionic conductivity properties of the SEBS membrane is improved by chemical modification and cross-linked S-SEBS-g-MA and may be suitable for applying RFB as a membrane in strong acid electrolyte. Sulfonated SEBS block copolymers have continuous ionic channels, while the AFM show direct evidence of well-ordered, nano-sized, and continuous ionic channels. The FT-IR result shows the sulfonic acid groups, carboxylic groups, and grafted MA being successfully introduced into SEBS. Our ion conductivity analysis revealed a significant improvement in the ionic conductivity from 0.1 to 0.25 S/cm, respectively for bare S-SEBS to CS-SEBS-g-MA membrane with increasing functional groups. Additionally, the ionic conductivity, water uptake and IEC of the S-SEBS, SEBS-g-MA, S-SEBS-g-MA and CS-SEBS-g-MA membrane are higher than those of the commercial Nafion 117 membrane. Ionic conductivity of the ionic exchange membrane increases with increasing functional group concentration. These results indicate the modified membrane is a promising candidate for large-scale energy storage systems that can further explored in the future. By manufacturing anion exchange membrane based on non-fluoro polymers such as SEBS, it will be possible to get price competitiveness compared to Nafion 117. Moreover, SEBS-based copolymers can be synthesized by different functional groups, which facilitates the improved ionic conductivity. The lower mechanical properties than conventional Nafion 117 can be improved by blending with alternative polymers or adding crosslink agents. We believe that this method could open up alternative strategy and promising approaches for its further development.

## Figures and Tables

**Figure 1 polymers-13-01669-f001:**
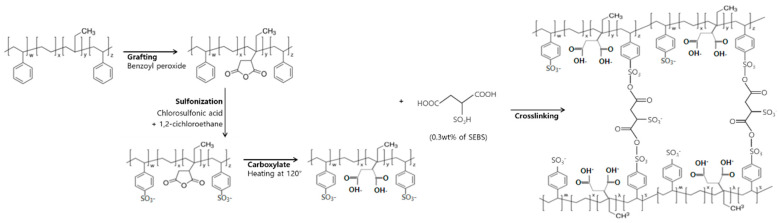
Scheme used for the preparation of MA-grafted SEBS based anion exchange membrane.

**Figure 2 polymers-13-01669-f002:**
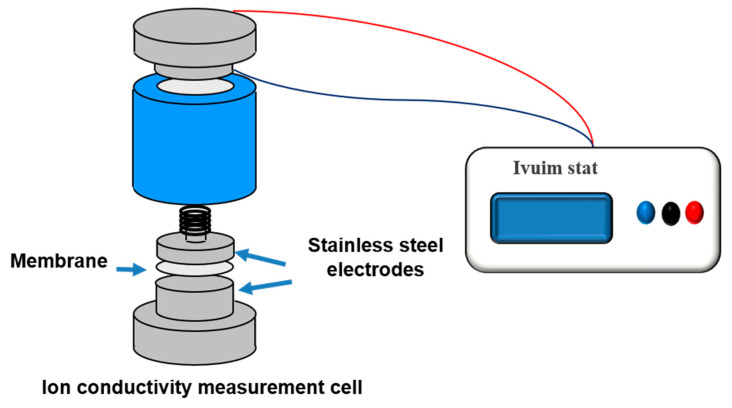
Experimental set-up for the ionic conductivity measurements of the different membranes.

**Figure 3 polymers-13-01669-f003:**
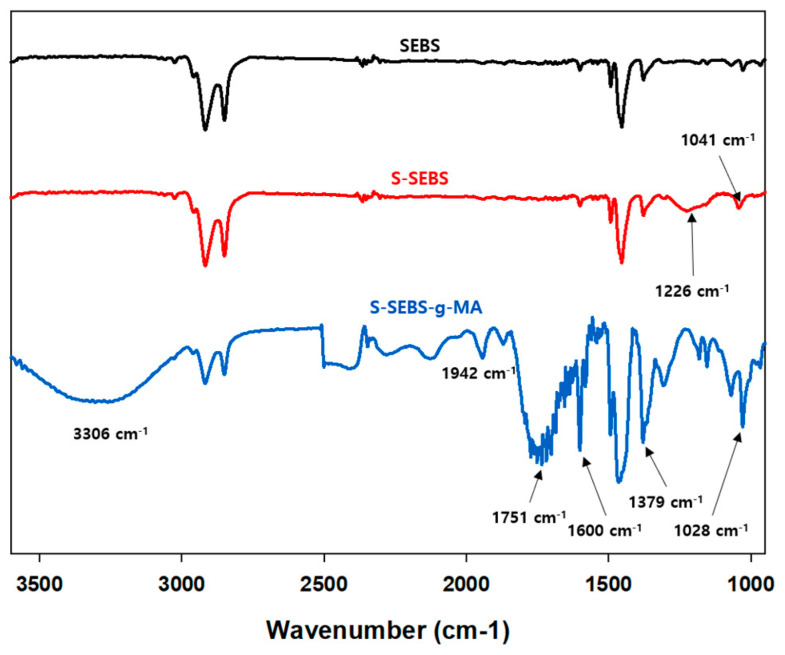
The FT-IR spectra of the SBES, S-SEBS, and S-SEBS-g-MA membranes.

**Figure 4 polymers-13-01669-f004:**
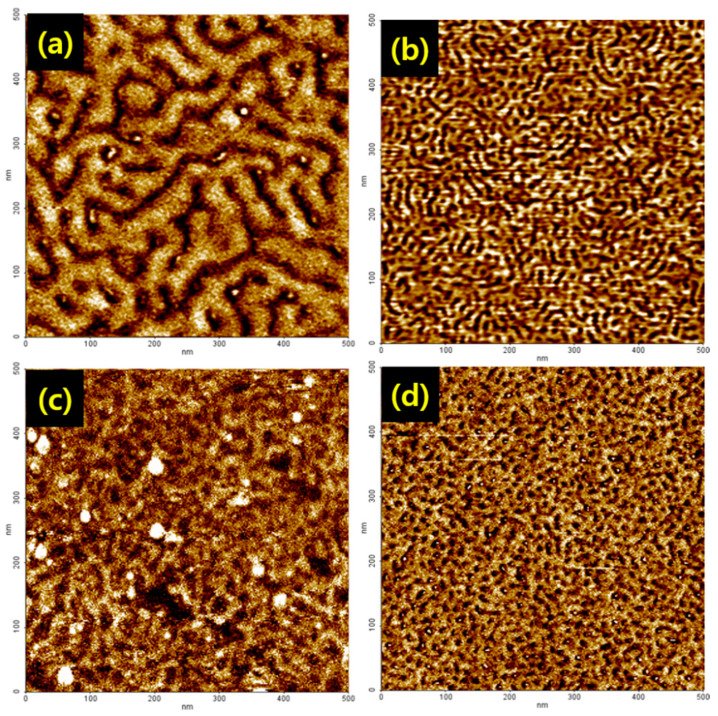
The AFM phase images of the modified membranes. (**a**) SEBS, (**b**) SEBS-g-MA, (**c**) S-SEBS, (**d**) S-SEBS-g-MA.

**Figure 5 polymers-13-01669-f005:**
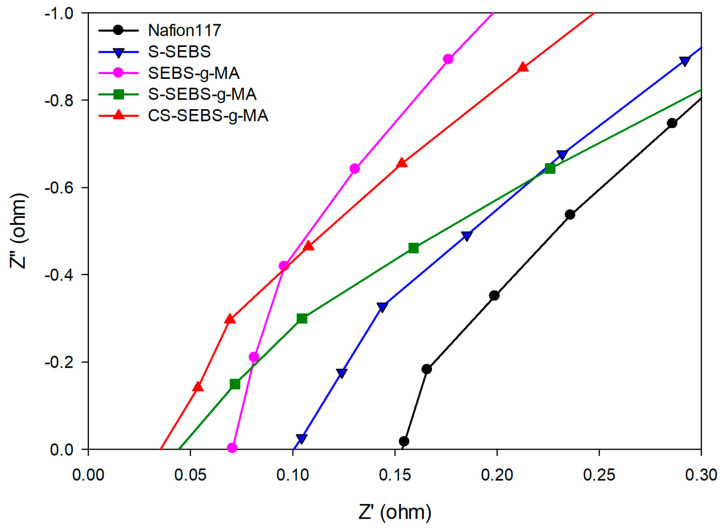
Impedance spectra of the Nafion 117, S-SEBS, SEBS-g-MA, S-SEBS-g-MA, and CS-SEBS-g-MA membranes. (Impedance data recorded at least for five sample from each set.).

**Figure 6 polymers-13-01669-f006:**
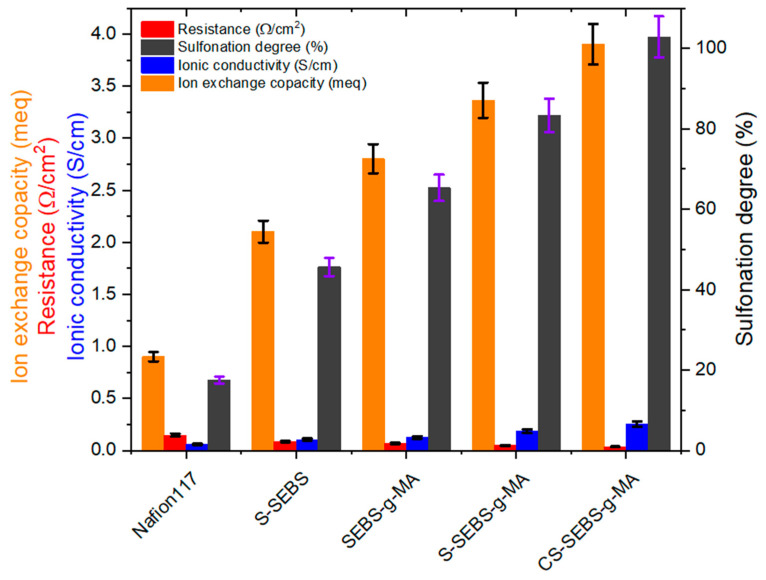
Degree of sulfonation, ionic conductivity and IEC of the different membranes studied in this work.

**Figure 7 polymers-13-01669-f007:**
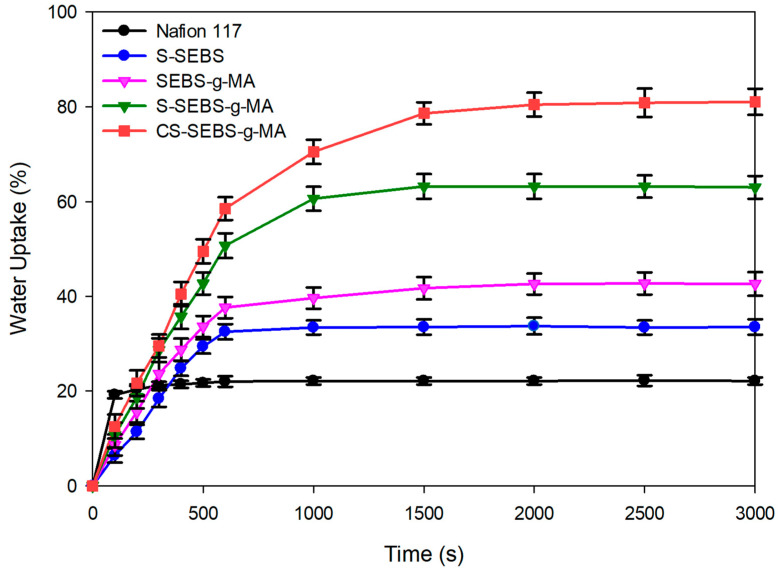
Water uptake of the Nafion 117, S-SEBS, SEBS-g-MA, S-SEBS-g-MA, and the CS-SEBS-g-MA membranes as a function of time. Water uptake data were recorded for at least five sample from each sample set.

**Figure 8 polymers-13-01669-f008:**
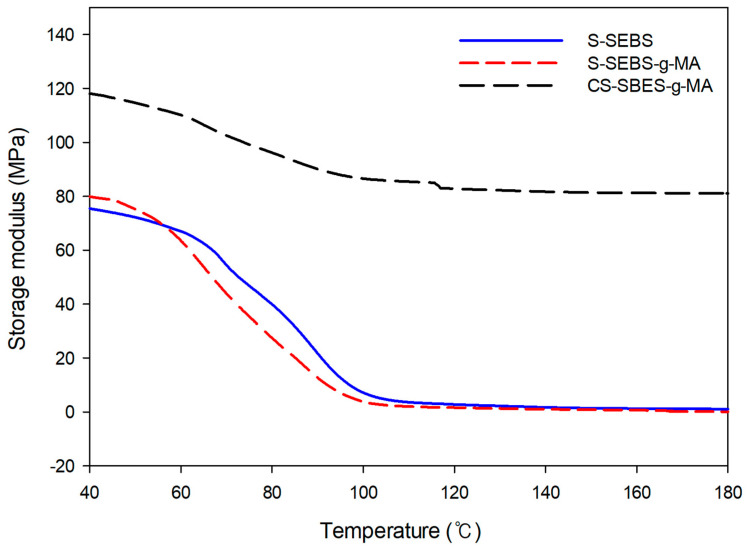
Storage modulus of the S-SEBS, S-SEBS-g-MA and CS-SEBS-g-MA.

**Table 1 polymers-13-01669-t001:** Average of M_w_ S-SEBS, SEBS-g-MA, S-SEBS-g-MA, and CS-SEBS-g-MA membranes.

Samples	SEBS	S-SEBS	SEBS-g-MA	S-SEEB-g-MA	CS-SEBS-g-MA
M_w_ (g/mol)	108,589	108,493	110,179	109,986	114,241

**Table 2 polymers-13-01669-t002:** Degree of sulfonation, ionic conductivity and IEC values obtained for the bare and modified membranes at room temperature.

Samples Name	Resistance (Ω/cm^2^)	Sulfonation Degree (%)	Ionic Conductivity (S/cm)	IEC (meq)
Nafion 117	0.145 ± 0.002	38.8 ± 0.4	0.06 ± 0.001	0.9 ± 0.002
S-SEBS	0.085 ± 0.0001	45.6 ± 0.1	0.1 ± 0.004	2.1 ± 0.004
SEBS-g-MA	0.06 ± 0.008	65.2 ± 0.1	0.12 ± 0.004	2.8 ± 0.004
S-SEBS-g-MA	0.047 ± 0.003	83.1 ± 0.2	0.18 ± 0.004	3.36 ± 0.06
CS-SEBS-g-MA	0.037 ± 0.002	102.6 ± 0.2	0.25 ± 0.004	3.9 ± 0.06

## Data Availability

Not applicable.

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
