# Peer review of "Anion Exchange Membrane Based on Sulfonated Poly (Styrene-Ethylene-Butylene-Styrene) Copolymers"

_polymers, 2021, doi:10.3390/polym13101669_

Round 1

Reviewer 1 Report

The use of renewable energy sources, such as solar and wind, is currently growing significantly owing to their environmental, social and economic benefits. However, these sources are intermittent and often unpredictable due to their dependency on the weather conditions. These characteristics limit the degree to which utilities can rely upon them, and currently such renewable energy alternatives comprise a small percentage of the primary power sources on the electrical grid. In this paper, Sulfonated poly styrene-ethylene-butylene-styrene copolymers (S-SEBS) was prepared, as anion exchange membranes, using casting method, and then modified with sulfonic acid groups and grafted with maleic anhydride (MA), to improve the ionic conducting properties. The prepared MA-grafted S-SEBS (S-SEBS-g-MA) membranes were characterized by Fourier transform infrared red (FT-IR) spectroscopy and dynamic modulus analysis (DMA). I am pleased to send you moderate comments. The results and theme of this paper is quite interesting. The layout is clear and easy to understand. Generally, this manuscript makes fair impression and my recommendation is that it merits publication in this Journal, after the following major revision:

  1. The authors need to reorganize the current introduction, which normally consists of three parts at least: background, literature review, brief of the proposed work. The current one is nothing but a literature review. Why their work is important comparing to previous reports? I think this is essential to keep the interest of the reader.
  2. The electrochemical analysis revealed the S-SEBS-g-MA anion exchange membrane showed ionic conductivity of 0.25 S/cm at 100% relative humidity, 72.5% water uptake capacity. The authors should give some explanation on above results and data.
  3. In Fig. 5 and 7, the authors should give the explanations for the difference of data collected from different sources.
  4. Materials and Methods part. Although the results look “making sense”, the current form reads like a simple lab report. The authors should dig deeper in the results by presenting some in-depth discussion.
  5. The use of renewable energy sources, such as proton exchange membrane fuel cells, is currently growing significantly owing to their environmental, social and economic benefits, see [International Journal of Hydrogen Energy, 2018, 43(37):17880-17888; Fractals, 2019, 27(2):1950012]. The exchange membrane has been widely used in many fields of life. For example, proton exchange membrane fuel cell (PEMFC) which includes a membrane, and two electrodes, has grown up with huge attraction because of its simple operation and fuel availability. Proton exchange membrane fuel cells have attracted attention from energy devices such as portable, mobile and stationary devices, since it helps effective reductions of energy shortage and environment pollution. Authors should introduce some related knowledge to readers.
  6. Please, expand the conclusions in relation to the specific goals and the future work.

Author Response

Response to Referee’s Comments

Manuscript number: Polymers-1158306 

Journal: Polymers

Title: Polymer blends based on sulfonated poly styrene-ethylene-butylene-styrene copolymers

First and foremost, authors would like to thank the Editor and Referee for valuable and critical comments to improve the quality of our paper substantially. The paper has been thoroughly revised and changes made in the paper are marked in RED and underlined.

Reviewers' comments:

Reviewer #1:

Reviewer #1: The use of renewable energy sources, such as solar and wind, is currently growing significantly owing to their environmental, social and economic benefits. However, these sources are intermittent and often unpredictable due to their dependency on the weather conditions. These characteristics limit the degree to which utilities can rely upon them, and currently such renewable energy alternatives comprise a small percentage of the primary power sources on the electrical grid. In this paper, Sulfonated poly styrene-ethylene-butylene-styrene copolymers (S-SEBS) was prepared, as anion exchange membranes, using casting method, and then modified with sulfonic acid groups and grafted with maleic anhydride (MA), to improve the ionic conducting properties. The prepared MA-grafted S-SEBS (S-SEBS-g-MA) membranes were characterized by Fourier transform infrared red (FT-IR) spectroscopy and dynamic modulus analysis (DMA). I am pleased to send you moderate comments. The results and theme of this paper is quite interesting. The layout is clear and easy to understand. Generally, this manuscript makes fair impression and my recommendation is that it merits publication in this Journal, after the following major revision:

Answer: Authors are very much thankful to reviewer’s positive comments and acceptance of our manuscript. Authors have checked carefully WHOLE manuscript and removed all grammatical mistakes. We hope the revised manuscript is now suitable for publication in “Polymers”.

  1. The authors need to reorganize the current introduction, which normally consists of three parts at least: background, literature review, brief of the proposed work. The current one is nothing but a literature review. Why their work is important comparing to previous reports? I think this is essential to keep the interest of the reader.

Answer: Authors are thankful to reviewer’s suggestions. Now, we have added the relevant literature, interest of this research work, content about why our work is important comparing to pervious report.

While many studies have been conducted on ionic exchange membranes with sulfonization of SESB, the ionic conductivity behavior between sulfonyl and carboxyl functional group in the ionic exchange membrane has not been carried out in previous studies. In this study, we would like to investigate the ionic conductivity of the SEBS membrane with the help of different behavior sulfonyl and carboxylic groups. We synthesized a high quality sulfonated SEBS-grafted-maleic anhydride membrane, which permits higher conductivity than conventional Nafion 117, and exhibits a good mechanochemical property with the help of sulfosuccinic acid by cross-linking mechanism. Our electrochemical analysis revealed that the modified membrane shows improved proton conductivity, ion exchange capacity (IEC) and water uptake properties. In addition, the dynamic modulus analysis (DMA) result confirms the improved modulus properties.

  1. The electrochemical analysis revealed the S-SEBS-g-MA anion exchange membrane showed ionic conductivity of 0.25 S/cm at 100% relative humidity, 72.5% water uptake capacity. The authors should give some explanation on above results and data.

Answer: Following the results between ionic conductivity and water uptake, we can expect that the investigating the functional groups such as sulfonyl and carboxylic acid of the membrane leads to increased ion conductivity. This gives rise to how ion conductivity is affecting on water uptake behavior of the membrane. Increasing functional group is directly proportional to the increasing the ability of the H+ transport in the membrane. This leads to improved ionic conductivity and water uptake.  

  1. In Fig. 5 and 7, the authors should give the explanations for the difference of data collected from different sources.

Answer: Authors are thankful to reviewer’s suggestions. We didn’t mention how many times we measure for each sample. However, we measured this data at least 5 times for each sample. We got identical data for each set. 

  1. Materials and Methods part. Although the results look “making sense”, the current form reads like a simple lab report. The authors should dig deeper in the results by presenting some in-depth discussion.

Answer: Authors are very much thankful to reviewer’s suggestions. We have discussed the detailed insight of the results and we believe that revised manuscript will add valuable literature in SEBS polymer membrane.  

  1. The use of renewable energy sources, such as proton exchange membrane fuel cells, is currently growing significantly owing to their environmental, social and economic benefits, see [International Journal of Hydrogen Energy, 2018, 43(37):17880-17888; Fractals, 2019, 27(2):1950012]. The exchange membrane has been widely used in many fields of life. For example, proton exchange membrane fuel cell (PEMFC) which includes a membrane, and two electrodes, has grown up with huge attraction because of its simple operation and fuel availability. Proton exchange membrane fuel cells have attracted attention from energy devices such as portable, mobile and stationary devices, since it helps effective reductions of energy shortage and environment pollution. Authors should introduce some related knowledge to readers.

Answer: Authors are very much thankful to reviewer’s valuable suggestions and references. We have revised introduction part carefully and discussed the given references.

  1. Please, expand the conclusions in relation to the specific goals and the future work.

Answer: Authors summarized the conclusion in-depth. 

Authors are very much thankful to reviewer’s positive response and acceptance of our manuscript for publication in “Polymers”. We have revised manuscript carefully and we hope the revised manuscript is now suitable for publication in “Polymers”.

The present manuscript has been revised thoroughly as three reviewer’s comments and added discussion accordingly. We believe that, the revised manuscript is now suitable for publication in “Polymers”. 

Reviewer 2 Report

  1. Title is misleading. There is no blending experiment throughout the manuscript. Consider "Anion exchange membrane based on sulfonated poly styrene-ethylene-butylene-styrene copolymers".
  2. What is the molecular weight distribution of these polymers after the modifications? Please add the data. 
  3. How was the degree of sulfonization evaluated? The experimental detail is missing. 
  4. Figure 1 should also illustrate the crosslinking step. 
  5. Figure 5, 6, 7, and Table 1 are missing the evidence of replicates. Please address.
  6. The conclusion of this work seems obvious: "Ionic conductivity of the ionic exchange membrane increases with increasing functional group concentration." Is there any other significant observation from this study?

Author Response

Response to Referee’s Comments

Manuscript number: Polymers-1158306 

Journal: Polymers

Title: Polymer blends based on sulfonated poly styrene-ethylene-butylene-styrene copolymers

First and foremost, authors would like to thank the Editor and Referee for valuable and critical comments to improve the quality of our paper substantially. The paper has been thoroughly revised and changes made in the paper are marked in RED and underlined.

Reviewers' comments:

Reviewer #2:

  1. Title is misleading. There is no blending experiment throughout the manuscript. Consider "Anion exchange membrane based on sulfonated poly styrene-ethylene-butylene-styrene copolymers".

Answer: Authors are agree with the reviewer’s suggestions. We modified the title of this article as follows:

“Anion exchange membrane based on sulfonated poly (styrene-ethylene-butylene-styrene) copolymers”.

  1. What is the molecular weight distribution of these polymers after the modifications? Please add the data.

Answer: Authors are thankful to reviewer’s critical comment. The molecular wt. distribution of the polymer would be the good for in-depth analysis. Unfortunately, currently this analysis is out-off our reach. However, in future we will undertake take this task.

  1. How was the degree of sulfonization evaluated? The experimental detail is missing.

Answer: We have added the detailed calculations about the degree of sulfonization in the Section 2.3.5.

We have calculate the sulfonization degree (SD) from using IEC.

where Mp is the molecular weight of the non-functional polymer repeat unit (SEBS) and Mf is the molecular weight of the functional group (SO3H). The values for Mp and Mf  are 118,000 and 81, respectively. The SD is related to the actual content of sulfonated poly-styrene group of S-SEBS.

  1. Figure 1 should also illustrate the crosslinking step.

Answer: We added the crosslinking step in the Figure1.

Figure 1. Scheme used for the preparation of MA-grafted SEBS based anion exchange membrane.

  1. Figure 5, 6, 7, and Table 1 are missing the evidence of replicates. Please address.

Answer: Authors are thankful to reviewer’s suggestions. Initially, we did not mention how many times we measure for each sample. However, we measured this data at least 5 times for each sample. We got identical data for each set. 

  1. The conclusion of this work seems obvious: "Ionic conductivity of the ionic exchange membrane increases with increasing functional group concentration." Is there any other significant observation from this study? 

Answer: Authors are thankful to reviewer’s suggestions. We have revised the conclusion section carefully.

Authors are very much thankful to reviewer’s positive response and acceptance of our manuscript for publication in “Polymers”. We have revised manuscript carefully and we hope the revised manuscript is now suitable for publication in “Polymers”.

The present manuscript has been revised thoroughly as three reviewer’s comments and added discussion accordingly. We believe that, the revised manuscript is now suitable for publication in “Polymers”. 

Round 2

Reviewer 1 Report

The revised form simply ignore my comments made in the first round. This manuscript should be rejected for published in Polymers. However, if the authors are willing to make the substantial revisions according to my the first comments, I would be glad to re-review this manuscript.

Author Response

Response to Referee’s Comments

Manuscript number: Polymers-1158306 

Journal: Polymers

Title: Polymer blends based on sulfonated poly styrene-ethylene-butylene-styrene copolymers

First and foremost, authors would like to thank the Editor and Referee for valuable and critical comments to improve the quality of our paper substantially. The paper has been thoroughly revised and changes made in the paper are marked in BLUE and underlined.

Reviewer #1:

Reviewers' comments:

The revised form simply ignored my comments made in first round. This manuscript should be rejected for published in polymers. However, if the authors are willing to make the substantial revisions according to my the first comments, I would be glad to re-review this manuscript.

Author’s action: We apologies for this. Now we have provided detailed explanation regarding the importance of membrane in PEMFC and added valuable references in the revised manuscript.

Please check our corrected response as below.  

Reviewer #1: The use of renewable energy sources, such as solar and wind, is currently growing significantly owing to their environmental, social and economic benefits. However, these sources are intermittent and often unpredictable due to their dependency on the weather conditions. These characteristics limit the degree to which utilities can rely upon them, and currently such renewable energy alternatives comprise a small percentage of the primary power sources on the electrical grid. In this paper, Sulfonated poly styrene-ethylene-butylene-styrene copolymers (S-SEBS) was prepared, as anion exchange membranes, using casting method, and then modified with sulfonic acid groups and grafted with maleic anhydride (MA), to improve the ionic conducting properties. The prepared MA-grafted S-SEBS (S-SEBS-g-MA) membranes were characterized by Fourier transform infrared red (FT-IR) spectroscopy and dynamic modulus analysis (DMA). I am pleased to send you moderate comments. The results and theme of this paper is quite interesting. The layout is clear and easy to understand. Generally, this manuscript makes fair impression and my recommendation is that it merits publication in this Journal, after the following major revision:

Answer: Authors are very much thankful to reviewer’s positive comments and acceptance of our manuscript. Authors have checked WHOLE manuscript carefully and removed all grammatical mistakes. We hope the revised manuscript is now suitable for publication in “Polymers”.

  1. The authors need to reorganize the current introduction, which normally consists of three parts at least: background, literature review, brief of the proposed work. The current one is nothing but a literature review. Why their work is important comparing to previous reports? I think this is essential to keep the interest of the reader.

Answer: Authors are thankful to reviewer’s suggestions. Now, we have added the relevant literature, interest of this research work and content about why our work is important compared pervious report.

While many studies have been conducted on ionic exchange membranes with sulfonization of SESB, the ionic conductivity behavior between sulfonyl and carboxyl functional group in the ionic exchange membrane has not been carried out in previous studies. In this study, we investigated the ionic conductivity of the SEBS membrane with the help of different sulfonyl and carboxylic groups. We synthesized a high quality sulfonated SEBS-grafted-maleic anhydride membrane, which permits higher conductivity than conventional Nafion 117, and exhibits a good mechanochemical property with the help of sulfosuccinic acid by cross-linking mechanism. Our electrochemical analysis revealed that the modified membrane shows improved proton conductivity, ion exchange capacity (IEC) and water uptake properties. In addition, the dynamic modulus analysis (DMA) result confirms the improved modulus properties. We strongly believe that this manuscript will add important literature in the membrane study.

  1. The electrochemical analysis revealed the S-SEBS-g-MA anion exchange membrane showed ionic conductivity of 0.25 S/cm at 100% relative humidity, 72.5% water uptake capacity. The authors should give some explanation on above results and data.

Answer: Following the results between ionic conductivity and water uptake, we can expect that the investigating the functional groups such as sulfonyl and carboxylic acid of the membrane leads to increased ion conductivity. This is because of the mobility of ions in the water phases increases with increasing water content. The modified membrane CS-SEBS-g-MA with sulfonyl and carboxylic acid functional groups lead to improved charge density of the modified membrane. From conductivity analysis, we observed significant improvement in the ionic conductivity from 0.1 to 0.18 S/cm respectively for bare S-SEBS to S-SEBS-g-MA membrane with increasing functional groups. Interestingly, this conductivity reached up to 0.25 S/cm for CS-SEBS-g-MA membrane. This is clearly indicates that water uptake and ionic conductivity is completely depending up on the functional groups and cross-linking. This gives rise to how ion conductivity is affecting on water uptake behavior of the membrane. Increasing functional group is directly proportional to the increasing the ability of the H+ transport in the membrane which leads to improved ionic conductivity and water uptake.

  1. In Fig. 5 and 7, the authors should give the explanations for the difference of data collected from different sources.

Answer: Authors are thankful to reviewer’s suggestions. We have collected this data for multiple samples prepared from same compositions. We have revised previous data and added an error bars in Figure 6 and 7. In our opinion, adding an error bar for impedance spectra does not make sense therefore we are reporting as obtained data for most reproducible samples. Instead, we have added the error values of impedance in Table1. In case of collecting data from different sources, unfortunately we do not have access and other facility to conduct these experiments in other institute. However, we have repeated these experiments several times and take mean values in order to satisfy the reliability. We hope the reviewer will understand this situation.

  1. Materials and Methods part. Although the results look “making sense”, the current form reads like a simple lab report. The authors should dig deeper in the results by presenting some in-depth discussion.

Answer: Authors are very much thankful to reviewer’s suggestions. We have discussed the detailed insight of the results and we believe that revised manuscript will add valuable literature in SEBS polymer membrane research.

  1. The use of renewable energy sources, such as proton exchange membrane fuel cells, is currently growing significantly owing to their environmental, social and economic benefits, see [International Journal of Hydrogen Energy, 2018, 43(37):17880-17888; Fractals, 2019, 27(2):1950012]. The exchange membrane has been widely used in many fields of life. For example, proton exchange membrane fuel cell (PEMFC) which includes a membrane, and two electrodes, has grown up with huge attraction because of its simple operation and fuel availability. Proton exchange membrane fuel cells have attracted attention from energy devices such as portable, mobile and stationary devices, since it helps effective reductions of energy shortage and environment pollution. Authors should introduce some related knowledge to readers.

Answer: Authors are very much thankful to reviewer’s valuable suggestions and references. We have revised introduction part carefully and discussed the given references.

  1. Please, expand the conclusions in relation to the specific goals and the future work.

Answer: Authors summarized the conclusion in-depth.

In conclusion, we successfully prepared the ionic exchange membrane from SEBS via sulfonation, grafting MA and crosslinking. This study showed that the ionic conductivity properties of the SEBS membrane is improved by chemical modification and cross-linked S-SEBS-g-MA and may be suitable for applying RFB as a membrane in strong acid electrolyte. Sulfonated SEBS block copolymers have continuous ionic channels. While the AFM show direct evidence of well-ordered, nano-sized, and continuous ionic channels. The FT-IR result shows the sulfonic acid groups, carboxylic groups, and grafted MA being successfully introduced into SEBS. Additionally, the ionic conductivity, water uptake and IEC of the S-SEBS, SEBS-g-MA, S-SEBS-g-MA and CS-SEBS-g-MA membrane are higher than commercial Nafion 117 membrane. Ionic conductivity of the ionic exchange membrane increases with increasing functional group concentration. These results indicate the modified membrane is a promising candidate for large-scale energy storage systems that can further explored in the future. By manufacturing anion exchange membrane based on non-fluoro polymer such as SEBS, it will be possible to get price competitiveness compared to Nafion 117. Moreover, SEBS based copolymers can be synthesize by different functional groups which facilitates the improved ionic conductivity. The low mechanical properties than conventional Nafion 117 can be improve by blending with alternative polymers or adding crosslink agents. We believe that this strategy would open alternative lvost-effective and promising approach for its further development.

Authors are very much thankful to reviewer’s positive response and acceptance of our manuscript for publication inPolymers. We have revised manuscript carefully and we hope the revised manuscript is now suitable for publication in “Polymers”.

The present manuscript has been revised thoroughly as three reviewer’s comments and added discussion accordingly. We believe that, the revised manuscript is now suitable for publication in Polymers”.  

Prof. (Dr.) Chang Kook Hong

Advanced Chemical Engineering Department

Chonnam National University

Gwangju, Korea 500-757.

Fax: +82-62-530-1849,

Email: [email protected] (C. K. Hong)

Reviewer 2 Report

  1. The revised Figure 1 demonstrated the crosslinking reaction between SEBS and sulfosuccinic acid. Is there any reference that the authors can provide to support the reaction mechanism?
  2. The authors mentioned that "The molecular wt. distribution of the polymer would be the good for in-depth analysis. Unfortunately, currently this analysis is out-off our reach. However, in future we will undertake take this task. " Molecular distribution is one of the most fundamental characterization that shall be carried out for a synthetic polymer. Please provide the data. 
  3. "However, we measured this data at least 5 times for each sample. We got identical data for each set. " Could the authors label that each replicate resulted in identical data in the figure caption, e.g. Figure 5, 6, 7, and Table 1?

Author Response

Response to Referee’s Comments

Manuscript number: Polymers-1158306 

Journal: Polymers

Title: Polymer blends based on sulfonated poly styrene-ethylene-butylene-styrene copolymers

First and foremost, authors would like to thank the Editor and Referee for valuable and critical comments to improve the quality of our paper substantially. The paper has been thoroughly revised and changes made in the paper are marked in BLUE and underlined.

Reviewers' comments:

Reviewer #2:

  1. The revised Figure 1 demonstrated the crosslinking reaction between SEBS and sulfosuccinic acid. Is there any reference that the authors can provide to support the reaction mechanism?

Answer: Authors added related references for this crosslinking reaction between SEBS and sulfosuccinic acid.

[1] G. M. Barrera, H. Lopez, V. M. Castano, R. Rodriuez, Studies on the rubber phase stability in gamma irradiated polystyrene-SBR blends by using FT-IR and Raman spectroscopy, Radiat. Phys. Chem, 2004, 69, 155-162

[2] J. A. Kerres, Development of ionomer membranes for fuel cells, J. Membr. Sci, 2001, 185, 3-27

  1. The authors mentioned that “the molecular wt. distribution of the polymer would be the good for in-depth analysis. Unfortunately, currently this analysis is out-off our reach. However, in future we will undertake take this task. “molecular distribution is one of the most fundamental characterization that shall be carried out for a synthetic polymer. Please provide the data.

Answer: Authors are thankful to reviewer’s critical comment.  We agree that this characterization can be analyzed by ssNMR. Unfortunately, we tried to get these data, but due to some restrictions, we are unable to provide. Authors are apologies for this deficiency. 

  1. “However, we measured this data at least 5 times for each sample. We got identical data for each set.” Could the authors label that each replicate resulted in identical data in the figure caption, e.g. Figure 5, 6, 7 and table1?

Answer: Authors are thankful to reviewer’s suggestions. We have collected this data for multiple samples prepared from same compositions. We have revised previous data and added an error bars in Figure 6 and 7. In our opinion, adding an error bar for impedance spectra does not make sense therefore we are reporting as obtained data for most reproducible samples. Instead, we have added the error values of impedance in Table1. In addition, we have repeated these experiments several times and take mean values in order to satisfy the reliability.

Authors are very much thankful to reviewer’s positive response and acceptance of our manuscript for publication in “Polymers”. We have revised manuscript carefully and we hope the revised manuscript is now suitable for publication in “Polymers.

The present manuscript has been revised thoroughly as per both reviewer’s comments and added discussion accordingly. We believe that, the revised manuscript is now suitable for publication in Polymers”.  

Prof. (Dr.) Chang Kook Hong

Advanced Chemical Engineering Department

Chonnam National University

Gwangju, Korea 500-757.

Fax: +82-62-530-1849,

Email: [email protected] (C. K. Hong)

Round 3

Reviewer 1 Report

In Ref. 1, volume “4” should be corrected as “43”

Author Response

Response to Referee’s Comments

Manuscript number: Polymers-1158306 

Journal: Polymers

Title: Polymer blends based on sulfonated poly styrene-ethylene-butylene-styrene copolymers

First and foremost, authors would like to thank the Editor and Referee for valuable and critical comments to improve the quality of our paper substantially. The paper has been thoroughly revised and changes made in the paper are marked in GREEN and underlined.

Reviewers' comments:

Reviewer #1:

  1. In Ref. 1, volume “4” should be corrected as “43”

Answer: We apologies for this. Authors revised the content you pointed out as below.

  1. Liang, Y. Liu, B. Xiao, S. Yang, Z. Wang, H. Han, An analytical model for the transverse permeability of gas diffusion layer with electrical double layer effects in proton exchange membrane fuel cells, Int. J. Hydrog. Energy, 2018, 43, 17880-17888.

Authors are very much thankful to reviewer’s positive response and acceptance of our manuscript for publication inPolymers. We have revised manuscript carefully and we hope the revised manuscript is now suitable for publication in “Polymers”.

The present manuscript has been revised thoroughly as three reviewer’s comments and added discussion accordingly. We believe that, the revised manuscript is now suitable for publication in Polymers”.  

Prof. (Dr.) Chang Kook Hong

Advanced Chemical Engineering Department

Chonnam National University

Gwangju, Korea 500-757.

Fax: +82-62-530-1849

Email: [email protected] (C. K. Hong)

Reviewer 2 Report

The authors are highly suggested to provide the SEC data for the molecular weight distribution of these polymers. 

Author Response

Response to Referee’s Comments

Manuscript number: Polymers-1158306 

Journal: Polymers

Title: Polymer blends based on sulfonated poly styrene-ethylene-butylene-styrene copolymers

First and foremost, authors would like to thank the Editor and Referee for valuable and critical comments to improve the quality of our paper substantially. The paper has been thoroughly revised and changes made in the paper are marked in GREEN and underlined.

Reviewers' comments:

Reviewer #2:

  1. 1The authors are highly suggested to provide the SEC data for the molecular weight distribution of these polymers.

Answer: Authors are thankful to reviewer’s critical comment. We agree that this characterization can be analyzed by GPC. We added the experimental method and result data of GPC.

2.3.2. Gel Permeation Chromatography (GPC)

The measurements were conducted using Shodex KF-804, Shodex KF-802 and Shodex KF-801 column (Tosoh, HLC-8320GPC) at 40℃ with the eluent flow rate of 1ml/min and injection of 100μl. All samples were dissolved in tetrahydrofuran (THF) before measurements. Polystyrene standard was used as reference.

3.1. Vibrational and GPC Analysis

Changes in weight average molecular weight (Mw) distribution for SEBS, S-SEBS, SEBS-g-MA, S-SEBS-g-MA and cross-linked S-SEBS-g-MA are presented in Table 1. The Mw of SEBS-g-MA, where MA is grafted on SEBS, is slightly increased compared to SEBS. But after sulfonization (S-SEBS, S-SEBS-g-MA), Mw decreased despite the addition of sulfonization compared to the pre-sulfonizaiton samples (SEBS, SEBS-g-MA). From this result decomposition was occurred in the polymer chain during the sulfonization. CS-SEBS-g-MA have highest molecular weight because the higher molecular weight content of the test specimen increases, which is attributable to crosslinking of the polymer chain.

Table 1. Average of Mw S-SEBS, SEBS-g-MA, S-SEBS-g-MA, and cross-linked S-SEBS-g-MA membranes.

Samples

SEBS

S-SEBS

SEBS-g-MA

S-SEEB-g-MA

CS-SSEBS-g-MA

Mw (g/mol)

108,589

108,493

110,179

109,986

114,241

Authors are very much thankful to reviewer’s positive response and acceptance of our manuscript for publication in “Polymers”. We have revised manuscript carefully and we hope the revised manuscript is now suitable for publication in “Polymers.

The present manuscript has been revised thoroughly as per both reviewer’s comments and added discussion accordingly. We believe that, the revised manuscript is now suitable for publication in Polymers”.  

Prof. (Dr.) Chang Kook Hong

Advanced Chemical Engineering Department

Chonnam National University

Gwangju, Korea 500-757.

Fax: +82-62-530-1849,

Email: [email protected] (C. K. Hong)
